# Damage Diagnosis of Single-Layer Latticed Shell Based on Temperature-Induced Strain under Bayesian Framework

**DOI:** 10.3390/s22114251

**Published:** 2022-06-02

**Authors:** Jie Xu, Zhengyang Zhao, Qian Ma, Ming Liu, Giuseppe Lacidogna

**Affiliations:** 1Key Laboratory of Coast Civil Structure Safety, Tianjin University, Ministry of Education, Tianjin 300350, China; jxu@tju.edu.cn; 2School of Civil Engineering, Tianjin University, Tianjin 300350, China; 2021205329@tju.edu.cn (Z.Z.); ming_liu@tju.edu.cn (M.L.); 3Binhai Industrial Research Institute Ltd., Tianjin University, Tianjin 300350, China; 4China Resources Land Limited, Yuedong Company, Huizhou 516000, China; 5Department of Structural, Geotechnical and Building Engineering, Politecnico di Torino, 10129 Torino, Italy; giuseppe.lacidogna@polito.it

**Keywords:** non-destructive inspection, damage diagnosis, temperature effects, latticed shell structure, Bayesian, Markov Chain-Monte Carlo methods

## Abstract

Under the framework of Bayesian theory, a probabilistic method for damage diagnosis of latticed shell structures based on temperature-induced strain is proposed. First, a new damage diagnosis index is proposed based on the correlation between temperature-induced strain and structural parameters. Then, Markov Chain Monte Carlo is adopted to analyze the newly proposed diagnosis index, based on which the frequency distribution histogram for the posterior probability of the diagnosis index is obtained. Finally, the confidence interval of the damage diagnosis is determined by the posterior distribution of the initial state (baseline condition). The damage probability of the unknown state is also calculated. The proposed method was validated by applying it to a latticed shell structure with finite element developed, where the rod damage and bearing failure were diagnosed based on importance analysis and temperature sensitivity analysis of the rod. The analysis results show that the proposed method can successfully consider uncertainties in the strain response monitoring process and effectively diagnose the failure of important rods in radial and annular directions, as well as horizontal (*x*- and *y*-direction) bearings of the latticed shell structure.

## 1. Introduction

Space grid steel structures, e.g., latticed shells, are widely used in large-scale public buildings with significant importance towards socio-economic development, such as airports, stations, and stadiums. Structural Health Monitoring (SHM) can effectively ensure structural safety by analyzing the vibration and static response of sensors deployed on the structures in service. In recent years, various SHM systems have been successfully deployed in engineering practice to accurately detect early damage in space steel structures [1,2,3].

Based on different types of monitored data, structural damage diagnosis methods can be categorized into vibration-based diagnosis methods and static-based diagnosis methods. In vibration-based diagnosis methods, researchers typically analyze the time domain or frequency domain of structural vibration acceleration and dynamic displacement data, extract the dynamic characteristics of structure, and combine with statistics or other methods to identify and locate the damage state [4,5]. However, the dynamic characteristics, including the natural frequency, are not only related to the stiffness of the structure itself, but also highly susceptible to environmental factors, such as temperature [6]. In addition, the use of such methods in the long-term damage diagnosis of complex spatial steel structures, such as latticed shells, is conventionally limited. This is due to the insensitivity of structural dynamic characteristics to local damage of structures, the large amount of dynamic data storage, and the complexity of the analysis process [7]. By contrast, the static-based diagnostic method can identify the damage states of local key elements with small data storage and analysis processes by monitoring the static response of structures, such as stress and deformation. Thus, this method has attracted higher attention in recent years for the damage diagnosis of bridges and other structures [8]. The traditional static-based damage diagnosis method usually requires the application of static loads as excitations, based on which the structural state is evaluated by observing the static response of the structure [9]. This procedure can be difficult for space grid steel structures, such as latticed shells, since in these structures, external loads, such as a vehicle, and piled-up weight are more difficult to model compared with bridge structures. The application of static-based diagnostic methods can, thus, be limited.

Subject to the combined effects of changing seasons, diurnal cycle, and solar radiation [10], space steel structures, such as latticed shells, generally exhibit significant temperature effects and daily fluctuations in local static response, such as that stresses are highly correlated with temperature [3,11]. In recent years, damage diagnosis methods based on temperature-induced response have raised attention from many researchers. Murphy et al. [12] used temperature-induced strain and displacement to identify the parameters of bridge structures. Kromanis et al. [13] used temperature-induced strain to evaluate the safety state of structures. Han et al. [14] used temperature-induced strain to modify the structural finite element model and identified the damage state of structures. However, the above methods are all deterministic diagnostic methods. In engineering practice, the daily monitoring data of static response contain not only temperature-induced components, but also various uncertainties, possibly resulting from wind load and noise interference. Thus, the influence of these uncertainties should be systematically considered in the damage diagnosis.

To this end, a probabilistic method for damage diagnosis of mesh and shell structures based on temperature-induced strain is proposed under the framework of Bayesian theory. First, a new damage diagnosis index is proposed based on the correlation between temperature-induced strain and structural parameters. Then, Markov Chain Monte Carlo is adopted to analyze the newly proposed diagnosis index, based on which the frequency distribution histogram of the posterior probability of the diagnosis index is obtained. Finally, the confidence interval of the damage diagnosis is determined by the posterior distribution of the initial state (baseline condition), and the damage probability of the unknown state is also calculated. The original latticed shell design of Beijing Laoshan Velo-drome was used as a case study to develop a finite element model, based on which the proposed method was applied and validated. The rod damage and bearing failure were diagnosed by using the proposed method based on the importance analysis and temperature sensitivity analysis of the rod to be monitored.

## 2. Damage Diagnosis Index and Monitoring Principle

### 2.1. Damage Diagnosis Index Based on Temperature-Induced Strain

As shown in Figure 1, a two-bar model is used to illustrate the basic principle of damage diagnosis based on temperature-induced strain, where ***a*** and ***b*** rods have the same material properties; specifically, they have modulus of elasticity *E*, coefficient of linear expansion *α*, and length *L*_0_. The cross-sectional areas of ***a*** and ***b*** rods are *A*_1_ and *A*_2_, respectively. In addition, the angle between the rod and the horizontal direction is *θ*, and the axial spring stiffness at the joint is *K_L_*. Typically, the temperature variation is non-uniform when rods are at different locations in the space structure. Therefore, the temperature variation of rods ***a*** and ***b*** is assumed to be Δ*T* and *β*Δ*T* in a certain time, respectively.

The strain transducer is assumed to be deployed in the axial direction of rod ***a***. Based on the structural mechanics theory, the temperature-induced strain response of rod ***a*** can be easily obtained as follows:(1)$$\varepsilon_{s}=-\frac{\alpha (1+\beta )\Delta T}{1+\frac{EA_{1}}{EA_{2}}+4\sin^{2}\theta \cdot \frac{EA_{1}}{K_{L}L0}}$$

Notably, $\varepsilon_{s}$ in Equation (1) refers to the constrained strain due to the constrained spring force, which is caused by the temperature change, i.e., the mechanical strain, rather than the free thermal strain of the structure. The damage diagnostic index *I_T_* is defined as the temperature-induced strain per unit temperature change, i.e.:(2)$$IT=\frac{\varepsilon_{s}}{\Delta T}$$

According to Equation (2), the relationship between the damage diagnosis index *I_T_* and the rod stiffness *EA*_1_, *EA*_2_, as well as the constraint stiffness coefficient *K_L,_* is plotted as shown in Figure 2. As can be seen from the figure, the damage diagnosis index *I_T_* is not only related to the measured rod stiffness (*EA*_1_), but also directly related to the adjacent rod stiffness (*EA*_2_) and the degree of the constraint of the rod (*K_L_*). Therefore, the change in *I_T_* can directly reflect the damage of critical elements in the structure, as well as the restraint changes in the supports, connected to the measured elements.

### 2.2. Principle of Simultaneous Monitoring of Temperature and Temperature-Induced Strain

In order to obtain damage diagnostic index *I_T_*, it is necessary to measure the structural surface temperature and axial strain simultaneously. Usually, these two types of data are collected by vibrating wire strain gauge [12] or fiber optic grating strain gauge [2]. For example, as can be seen from Figure 3a, this is a site photo of a vibrating wire strain gauge which have been installed on the surface of a space grid steel structure during construction, so that strain and temperature data, as shown in Figure 3b, can be collected simultaneously. The sensor mainly consists of two mounting blocks, protection tubes, wire, coils (excitation coils, receiving coils, thermistors) and cables (as shown in Figure 4). The strain gauge mounting blocks are used not only to hold the vibrating wire in place, but also to ensure the synchronous deformation of the strain gauge and the structure; the thermistor is used to synchronize the temperature data. First, when the structure under test is deformed, the strain gage measures the deformation simultaneously. Then, the deformation is transformed into stress changes in the wire through two mounting blocks, thus, changing the vibration frequency of the wire. Next, the electromagnetic coil excites the wire and measures its vibration frequency. Finally, the frequency signal is transmitted via cable to the acquisition device to measure the internal strain of the tested structure.

The specific temperature–strain measurement principle can be further explained in conjunction with Figure 4. In the initial condition without external forces, the wire vibrates with a steady amplitude according to the initial stress, whose working state meets the condition of soft undamped micro-vibration, so that the vibration frequency can be determined as follows [15],
(3)$$f_{0}=\frac{1}{2l_{0}}\sqrt{\frac{\sigma_{0}}{\rho }}$$
where *f*_0_ is the initial frequency; *l*_0_ is the initial effective length of the wire; *ρ* is the density of the wire; and *σ*_0_ is the initial stress of the wire.

When the deformation of the structure causes a corresponding stretching or compression of the strain gauge, the stress in the wire will increase or decrease. At this point, the initial frequency will also increase or decrease to *f*. Since the strain gauge has a limited measurement range, approximation of $l_{0}+\Delta l\approx l_{0}$ holds. Thus, the following can be obtained,
(4)$$f=\frac{1}{2l_{0}}\sqrt{\frac{\sigma_{\mathrm{VW}}}{\rho }}$$

Since the mass *m*, length *l*_0_, cross-sectional area *A*_VW_, as well as modulus of elasticity *E*_VW_ of the wire can be viewed as constants, the relationship between stress and output frequency of the wire can be regarded as follows,
(5)$$\sigma_{\mathrm{VW}}\propto f^{2}$$

The strain increment $\varepsilon_{VW}$ (output strain signal of the strain gauge) of the steel string can be calculated by the following formula
(6)$$\varepsilon_{\mathrm{VW}}=\kappa \cdot \left(f^{2}-f_{0}^{2}\right)$$
where κ is a constant.

From Section 2.1, the actual strain of the structure under temperature change in the elastic constrained state can be calculated by the mechanical strain as follows.
(7)$$\varepsilon =\alpha \Delta T-\varepsilon_{s}$$

Unlike in the case of ordinary external forces, the change in tension of the strain gauge steel chord under the action of temperature comes from two sources. Specifically, one source is the change in tension with the temperature deformation of the measured structure, i.e., $\Delta P_{1}=\left(E_{\mathrm{VW}}A_{\mathrm{VW}}\right)\cdot \varepsilon$, while the other is the change in tension of the steel string itself due to temperature deformation, i.e., $\Delta P_{2}=-\left(E_{\mathrm{VW}}A_{\mathrm{VW}}\right)\cdot \left(\alpha_{\mathrm{VW}}\Delta T\right)$. The actual tension of the steel string changes is shown in the following equation.
(8)$$\Delta P=\Delta P_{1}+\Delta P_{2}=E_{\mathrm{VW}}A_{\mathrm{VW}}\cdot \left(\alpha \Delta T-\varepsilon_{s}\right)-E_{\mathrm{VW}}A_{\mathrm{VW}}\cdot \left(\alpha_{\mathrm{VW}}\Delta T\right)$$

Therefore, under the effect of temperature, the relationship between the mechanical strain increment of the steel string and the output strain signal of the strain gauge $\varepsilon_{\mathrm{VW}}$ can be derived as follows.
(9)$$\varepsilon_{s}=\left(\alpha -\alpha_{\mathrm{VW}}\right)\Delta T-\varepsilon_{\mathrm{VW}}$$

According to Equation (9), the temperature-induced strain of the structure can be effectively obtained. However, in the actual health monitoring system, the static response monitoring data of the structure result from the combined effect of several factors. However, temperature is still one of the main factors affecting the structural response variation in the daily condition [6]. By preprocessing the data through principal component analysis [16], empirical mode decomposition [17], and blind source separation [18], the temperature-induced components can be effectively separated from the overall static response monitoring data, and then the damage diagnosis index can be constructed according to Equation (9) for further damage diagnosis of the structure.

## 3. Probabilistic Damage Diagnosis Based on Bayesian Theory

Theoretically, the safety state of critical structural rods and supports can be directly assessed in real time, which is possible based on the observation of changes in the I_T_ in Section 2. However, the temperature-induced strain of the structure is more complicated during the actual field monitoring case. In addition to the uniform temperature component, the temperature-induced strain may also be influenced by non-uniform temperature, as well as many uncertainties, such as noise. Therefore, a probabilistic damage diagnosis method based on Bayesian theory is proposed, which is sampled and updated using the measurements of damage diagnosis index mentioned above, whereby the effect of uncertainty on damage diagnosis is taken into account.

Specifically, for a given operating state M, the joint posterior probability distribution of the true value *I_T_* to be identified can be obtained by Bayesian formula under the condition that the measured data set $\tilde{I_{T}}$ of the structural damage diagnosis index is known [19],
(10)$$p\left(I_{T}|\tilde{I_{T}},M\right)=\frac{p\left(\tilde{I_{T}}|I_{T},M\right)p\left(I_{T}|\;M\right)}{p\left(\tilde{I_{T}}\;|\;M\right)}$$
where $p\left(\tilde{I_{T}}|I_{T},M\right)$ is the probability distribution of the damage diagnosis index measurement set $\tilde{I_{T}}$ (a subset of samples), often called the likelihood function, which is a function of *I_T_*; $p\left(I_{T}\;|\;M\right)$ is the prior probability distribution of the parameter vector $I_{T}\;$, generally based on engineering experience and obtained from historical data; in this paper, it is assumed to obey a normal distribution. This hypothesis of distribution is reasonable for the population [20], and its mean and variance are estimated from the mean and variance of $\tilde{I_{T}}$; $p\left(\tilde{I_{T}}\;|\;M\right)$ is a normalization constant, unrelated to $I_{T}$.

Since the posterior distribution of the true value $I_{T}$ is complex and non-standard, this paper uses the MCMC method to approximate the calculation. The basic idea of the MCMC method is to obtain samples of smoothly distributed Markov chains, and then make statistical inferences based on these extracted samples [20]. The Metropolis–Hastings (MH) algorithm is a frequent sampling method of the MCMC [21]. In this paper, the MH algorithm is used to calculate the above posterior distribution, i.e., sampling from the measurement set $\tilde{I_{T}}$, which starts from the initial values, and then an integrable non-periodic Markov chain is obtained according to the proposed distribution. The probability density function curve of this Markov chain is the posterior probability distribution of the diagnostic indicators considering uncertainty.

After obtaining the posterior distribution of the initial state (baseline case), the upper (or lower) confidence limit of its 95% guarantee is further determined. The damage probability of the unknown state is calculated by confidence intervals, where the sum of the relative frequencies greater than the upper confidence limit (or less than the lower confidence limit) is defined as the damage probability (as shown in Figure 5). As the number of samples increases, the relative frequency distribution histogram approaches its fitted normal distribution probability density function. In this paper, in order to simplify the calculation process, the histogram of relative frequency distribution is used instead of the probability density curve for the damage probability calculation.

## 4. Single-Layer Spherical Mesh Shell Damage Diagnosis

### 4.1. Introduction of Engineering Background and Establishment of Finite Element Model

The finite element numerical simulation of a structure was carried out using ABAQUS to verify the effectiveness of the method in this paper. Specifically, the modeling was based on a rib ring tilt rod-type single-layer spherical mesh shell structure, which is the original roof design of Beijing Laoshan Velodrome [22]. The overall structure is shown in Figure 6. The vertical projection of the structure is circular, with a diameter of 147.2 m, a sagittal height of 14.7 m, and a structural height of 0.9 m. There are 24 radial rib beams, divided equally into eight sections along the ring direction, which are all H-beams with a specification of H900×300×16×18∼H900×300×16×18. Furthermore, the supports between the rib ring beams are round steel pipes, and their specifications are Φ400×10~Φ550×10. The horizontal projection of the outer ring beam truss is an isosceles triangle with a width of 8800 mm and height of 6800 mm, and the lower chord of the truss is Φ1000×18 round steel pipe. In addition, the radial rib beam support bar is a single-limb herringbone round steel tube with the specification of Φ1000×18.

For the numerical simulation, circumferential and radial rib beams, support diagonal bars and single-limb herringbone round steel pipe support columns are simulated by B31 unit, the material is Q345 steel, the modulus of elasticity is 2.06 GPa, Poisson’s ratio is 0.3, the density is 7850 kg·m^−3^, the coefficient of thermal expansion is 1.22 × 10^−5^, and the support is connected by a three-way hinge. The constant load is 1.1 kN·m^−2^ and the live load is 0.5 kN·m^−2^. A standard combination of the two was made and converted to equivalent nodal loads assigned to each node (as in Figure 7).

### 4.2. Monitoring Rod Selection and Temperature Working Condition Setting

Considering the symmetry of the structure, the shaded part of the sector shown in Figure 8 is selected as the area to be monitored. Due to the large number of bars in the structure, in order to effectively reflect the safety state of the structure with a limited number of sensors, it is necessary to select the key bars with high structural importance for key monitoring. For spatial grid structures, such as latticed shells, the importance of the bars is not only related to the stiffness of the bars, but also related to the force transmission path of the structure. Once the bars located on the main force transmission path of the structure are damaged, the structure may produce large deformation or even continuous collapse [23]. The amount of structural strain energy change due to member failure reflects the effect of the member on the overall performance and stiffness of the structure. The greater the structural strain energy change, the greater the role and influence of the member in the structure [24]. Therefore, this paper defines the bar importance factor as the relative rate of change in the total strain energy of the structure after the removal of a certain bar.
(11)$$\gamma_{m}=\frac{C_{d}-C_{0}}{C_{d}}$$
where *C*_0_ and *C_d_* are the strain energy of the structure before and after the removal of a certain bar, respectively. From Equation (11), the importance factor is directly related to the importance of the removed bar to the structure. The larger the importance coefficient, the greater the importance of the rod. The importance coefficient is then normalized as follows.
(12)$$\gamma_{m}^{*}=\frac{\gamma_{m}-\gamma_{m,\min}}{\gamma_{m,\max}-\gamma_{m,\min}}$$
where $\gamma_{m,\max}$ and $\gamma_{m,\min}$ are the largest and smallest importance factors among all bars, respectively. By the result of the calculation, some of the final determined rods with relatively large importance coefficients ($\gamma_{m}^{*}>0.2$) of the area to be monitored are shown as red marks in Figure 8.

In the actual monitoring process, the resolution of strain sensors is usually around 1με. If the strain on the rod is less than 1 με per unit temperature change, a significant error may exist in damage diagnosis progress when the proposed method is used. Therefore, further temperature sensitivity analysis was performed on the above-mentioned bars with larger importance factors. As a result, the bars with higher temperature sensitivity are selected for key monitoring. A uniform temperature field with unit temperature change (warming) is applied to the structure, and the strain response of each of the above-mentioned bars is shown in Figure 9. The temperature sensitivity of 97, 169, 313, 397, and 493 rods is low, which may cause large errors when using the method in this paper for damage diagnosis, while the temperature sensitivity of 1, 146, 170, 193, 373, 374, 421, and 469 rods is high, and using the method in this paper for damage diagnosis may obtain more satisfactory damage diagnosis results. In order to compare and illustrate the applicability of this paper’s method, this paper finally selected the highest temperature sensitivity of No. 1 circumferential rod (heating pressure), No. 146 radial rod (heating pressure), and No. 469 diagonal brace member for analysis, as shown by the green markers in the bar chart of Figure 9.

At the location of the rod to be monitored, sensors for simultaneous monitoring of temperature and strain data are placed along the axial direction of the rod to monitor the surface temperature and axial strain response of the structure. We assumed that the monitoring frequency of the sensor is 1 time·day^−1^; in order to make the temperature variation closer to the actual monitoring conditions, the possible non-uniform temperature effect of the structure caused by solar radiation during the day can be considered. The daily data collection moment is 0:00 at night. As such, 0:00 temperature variation is assumed to apply to the structure, as shown in Figure 10, which is a typical meteorological year in Beijing [6]. Meanwhile, considering the influence of noise and wind load on the data during the monitoring process, the diagnostic indexes obtained from numerical simulation are added with uncertainty interference according to the following formula:(13)$$\tilde{I_{T}}=I_{T}+\lambda \cdot \bar{I_{T}}\cdot N(0,\;1)$$
where $I_{T}$ and $\tilde{I_{T}}$ are the original numerical simulated value of the damage diagnostic index and the measured value after adding the uncertainty interference, respectively. N(0 , 1) is a Gaussian distributed random number with 0 as the mean and 1 as the variance. $\bar{I_{T}}$ is the mean value, *λ* is the uncertainty level, i.e., the relative average deviation calculated from the field monitoring data, which is taken as 0.2 in this paper.

### 4.3. Rod Damage Diagnosis

Damage simulations are performed on the important rods at the sensor deployment locations to verify the effectiveness of the method in this paper for rod damage diagnosis. In the actual monitoring process, there are many damage types in the structure. In this paper, the loss of stiffness of the unit due to damage is simulated by reducing the modulus of elasticity by 30%, 60%, and 90%, respectively, to simulate three degrees of damage to the rod: mild, moderate, and severe.

In accordance with the process in Section 3, the posterior probabilities of the damage diagnostic indicators described in this paper were estimated, and the relative frequency distributions of the posterior probabilities for different rods are shown in Figure 11, Figure 12 and Figure 13. It can be seen that the absolute value of the mean value of the diagnostic indexes of the rods tends to decrease as the degree of damage increases. Among them, the changes in the No.1 circumferential rod and No.146 radial rod are more obvious, and the damage at this location can be directly and qualitatively judged by the changes in the mean value of the diagnostic indexes.

Further, in order to make a more accurate diagnosis for the damage of the monitored members, the 95% upper and lower confidence limits were determined according to the data of each rod health condition (baseline condition), and the damage probabilities under different damage conditions were calculated as shown in Table 1. As shown in the table, even for minor damage, the damage probability of No. 1 and No. 146 can reach large values, and with the increase in the degree of damage, the probability of damage at both locations reached 100%, while for No. 469 diagonal spar, in its minor and moderate damage, the probability of damage is at a low level, and only when more serious damage occurs does the probability of damage reach a large value. Therefore, the early damage could not be effectively diagnosed by the method in this paper, which may be due to the fact that the degree of restraint of the supporting diagonal rod is weaker than that of the circumferential and radial rod, and the change in temperature-induced strain before and after the damage is smaller.

### 4.4. Bearing Restraint Failure Diagnosis

The effectiveness of the method is further verified in diagnosing the constraint failure of the support. Taking the bearing in the area to be monitored, as shown in Figure 14, as an example, the bearing restraint failure conditions in three directions are considered and simulated by removing the restraints in *x*, *y* and *z* directions, respectively. The temperature-induced strains of bars 1146 and 469 are monitored to diagnose each of the three bearing restraint failure conditions.

The relative frequency distributions of the posterior probabilities of different bars calculated according to the method of this paper are shown in Figure 15, Figure 16 and Figure 17. As shown in the figures, both the 1146 and 469 rods are most sensitive to the *x*-directional bearing restraint failure, where the 469 diagonal spar has the most obvious change in the mean relative frequency after the *x*-directional bearing restraint failure, so the type of bearing damage can be judged qualitatively by the diagonal spar; after the *y*-directional and *z*-directional bearing restraint failure, the mean change in the diagnostic index is smaller, and further damage needs to be performed (probability analysis).

The upper and lower confidence limits of the health conditions to determine the abnormal probability of each rod under different support constraint failure conditions are shown in Table 2. As shown in the table, the damage probability of 1146 and 469 for *x*-direction bearing restraint failure reaches 100%; the *y*-direction bearing restraint failure also has a high abnormal probability, while for *z*-direction bearing restraint failure, the damage probability is low, which cannot effectively diagnose such damage.

## 5. Conclusions

In this paper, a damage diagnosis method based on temperature-induced strain for a single-layer latticed shell structure is proposed under the framework of Bayesian theory. The method uses the temperature-induced strain data measured by sensors deployed in the key monitoring area of the structure and the ambient temperature data to construct a new damage diagnosis index; the posterior probability distribution of the damage diagnosis index is estimated by the MCMC method; the confidence interval of the damage diagnosis is determined according to the initial state (baseline condition), and then the damage of the unknown state of the structure is diagnosed by the calculation of the damage probability. The effectiveness of the method is verified by the numerical simulation of a mesh-shell structure, and the following conclusions are drawn:The temperature-induced strain is directly related to the rod stiffness and the restraint stiffness of the structure. The damage diagnosis indexes constructed by the temperature-induced response can theoretically diagnose the damage to critical members of the structure and the restraint failure of the support.By performing importance analysis and temperature sensitivity analysis on the bars, the bars with high importance to the overall safety of the structure and high-temperature sensitivity can be effectively selected as the key monitoring bars from the many monitoring bars.In practical application, the method in this paper is more effective in diagnosing the damage of annular and radially important bars, as well as the failure of horizontal direction support restraint, and less effective in diagnosing the damage of oblique spars, as well as the failure of *z*-direction support restraints.The validity of the method proposed has been initially verified based on the finite element simulation. However, in the field monitoring environment, the measured data are much more complicated than the numerical simulation. In the next work, the effectiveness of the method will be further verified in the application of actual monitoring projects.

## Figures and Tables

**Figure 1 sensors-22-04251-f001:**
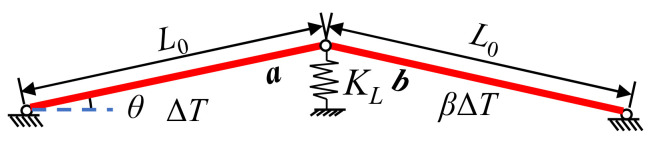
Schematic diagram of the temperature change in the two-pole structure.

**Figure 2 sensors-22-04251-f002:**
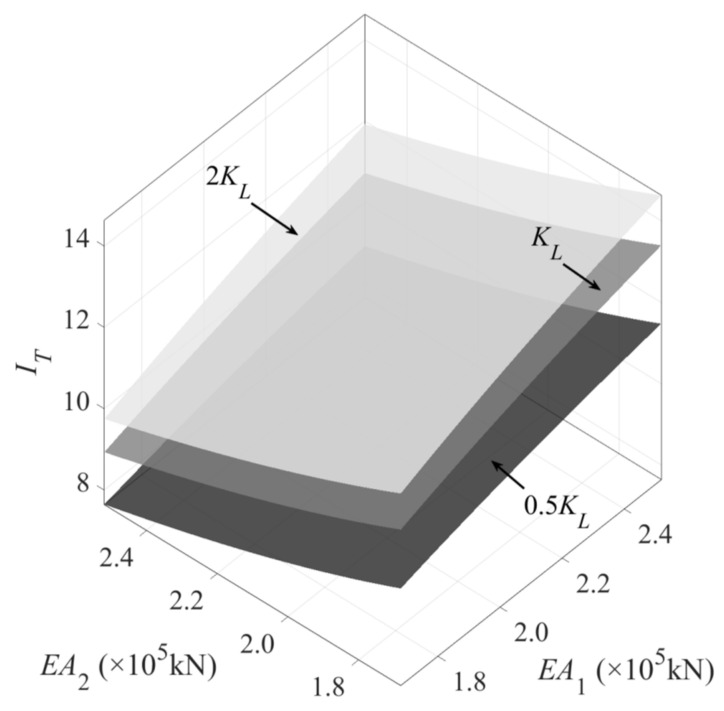
The relationship between damage diagnosis indicators and structural parameters.

**Figure 3 sensors-22-04251-f003:**
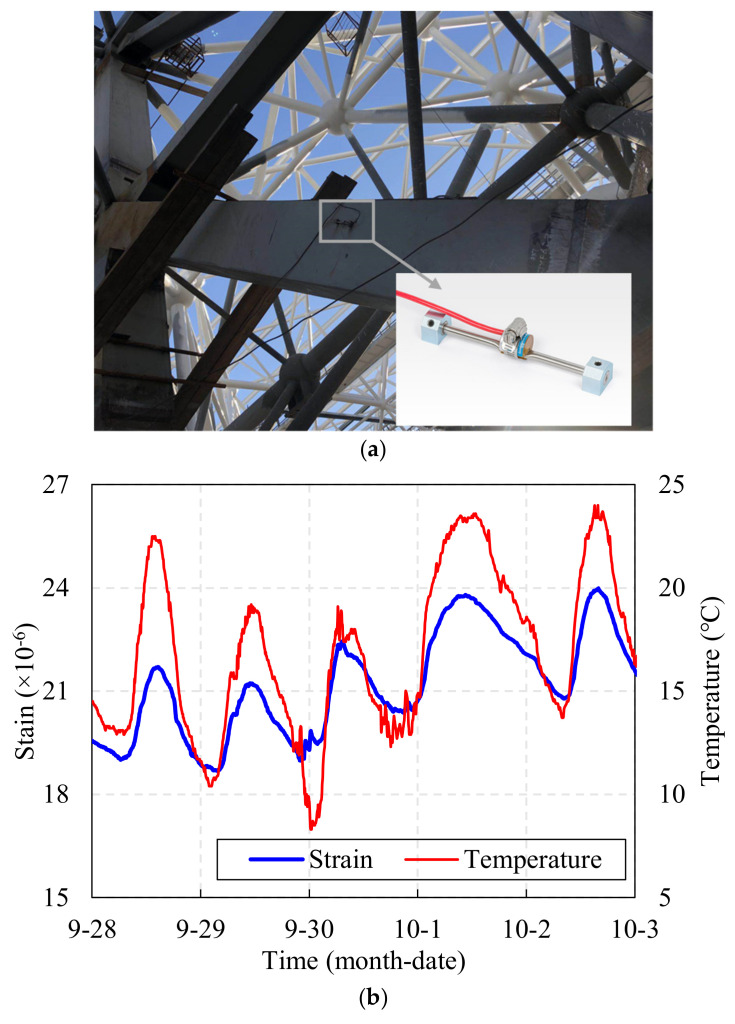
Arc weldable vibrating wire strain gauge of a space grid steel structure. (**a**) Photos of the on-site installation, (**b**) temperature and strain data for a period.

**Figure 4 sensors-22-04251-f004:**
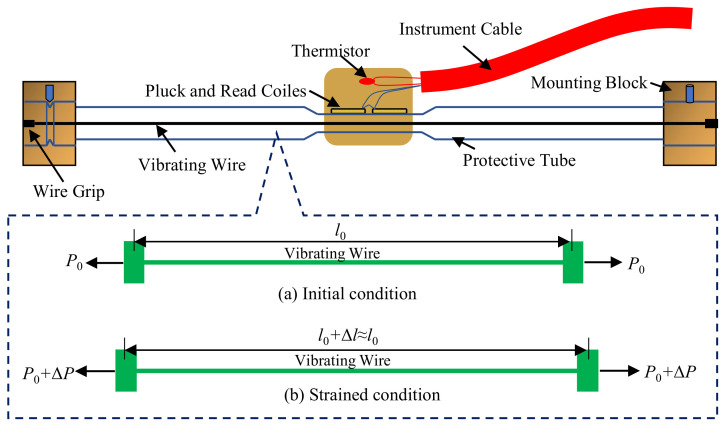
Schematic diagram of the internal details and measurement principle of the vibrating wire strain gauge.

**Figure 5 sensors-22-04251-f005:**
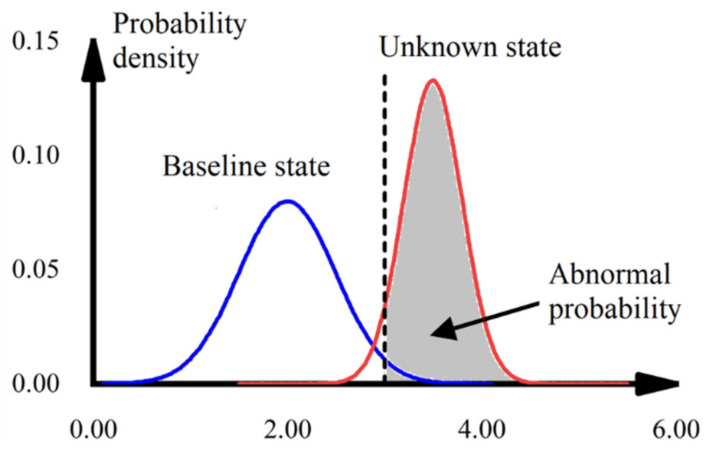
Schematic diagram of damage probability calculation.

**Figure 6 sensors-22-04251-f006:**
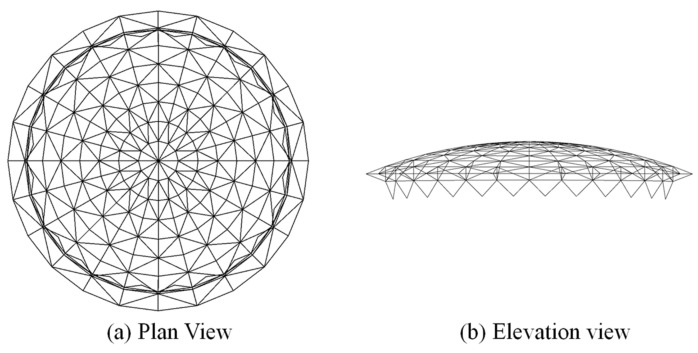
Overall structure diagram.

**Figure 7 sensors-22-04251-f007:**
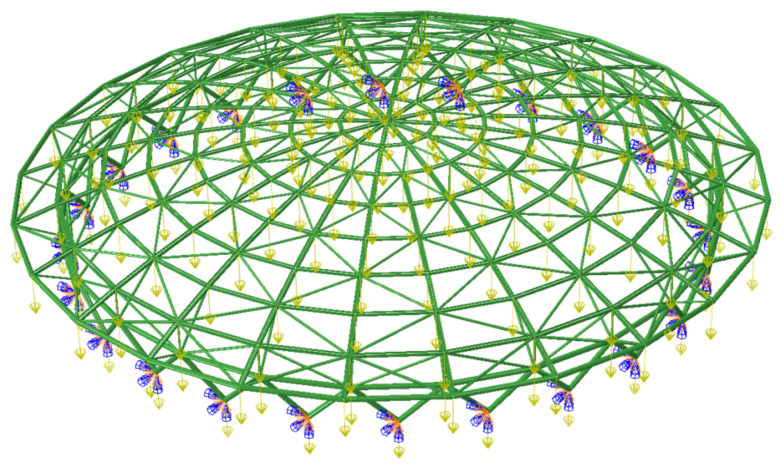
ABAQUS finite element model.

**Figure 8 sensors-22-04251-f008:**
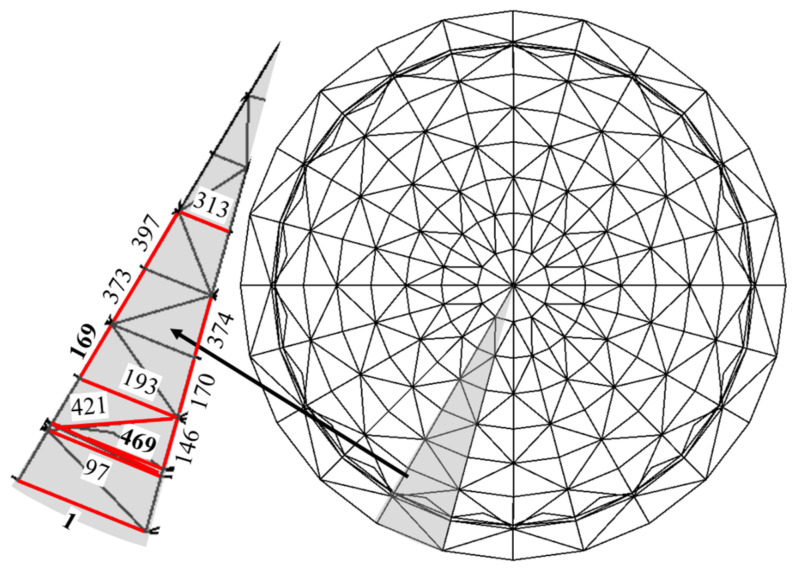
The monitored area and the location of important members.

**Figure 9 sensors-22-04251-f009:**
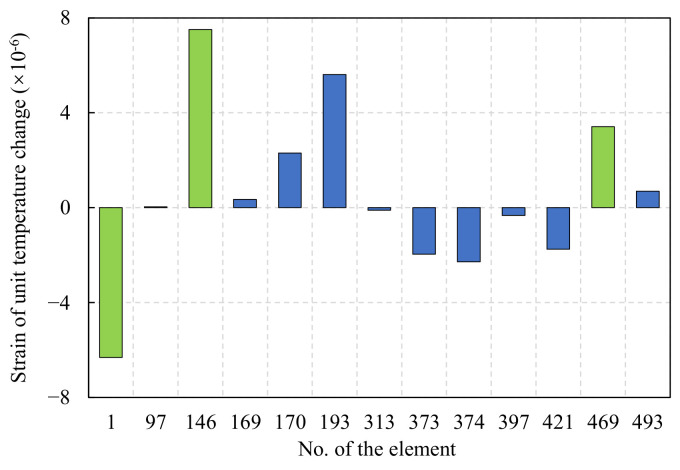
Strain under unit temperature increase of important member.

**Figure 10 sensors-22-04251-f010:**
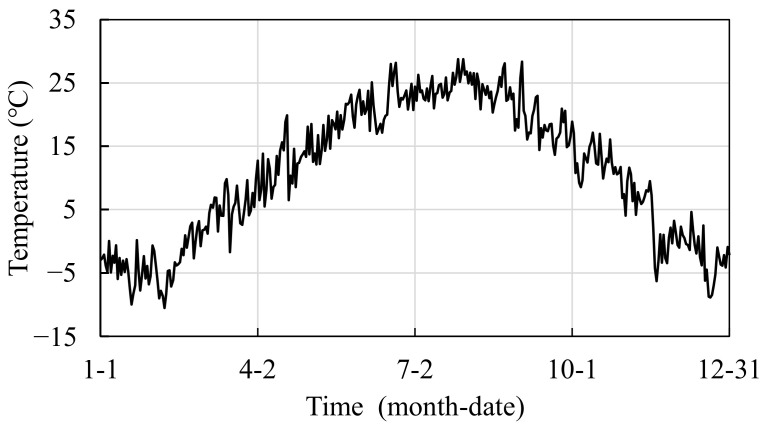
The average daily temperature of a place in a year.

**Figure 11 sensors-22-04251-f011:**
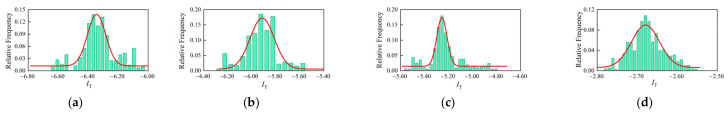
Posteriori relative frequency distribution of member 1. (**a**) Healthy; (**b**) mild injury; (**c**) moderate damage; (**d**) severe injury.

**Figure 12 sensors-22-04251-f012:**
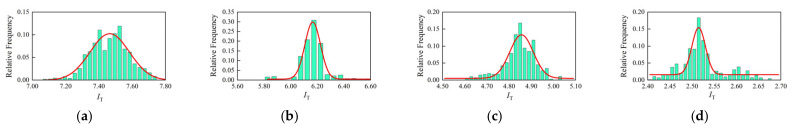
Posteriori relative frequency distribution of member 146. (**a**) Healthy; (**b**) mild injury; (**c**) moderate damage; (**d**) severe injury.

**Figure 13 sensors-22-04251-f013:**
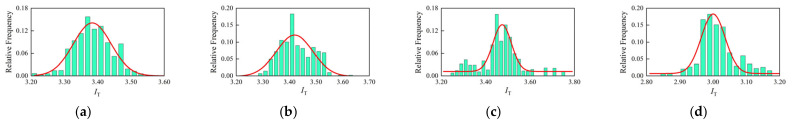
Posteriori relative frequency distribution of member 469. (**a**) Healthy; (**b**) mild injury; (**c**) moderate damage; (**d**) severe injury.

**Figure 14 sensors-22-04251-f014:**
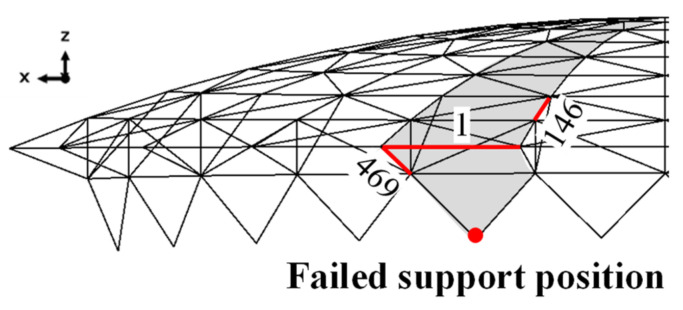
Failed supports and monitored critical member locations.

**Figure 15 sensors-22-04251-f015:**
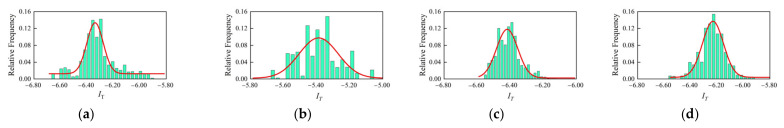
Posteriori relative frequency distribution of member 1. (**a**) Healthy; (**b**) failure of *x*-directional support restraint; (**c**) failure of *y*-directional support restraint; (**d**) failure of *z*-directional support restraint.

**Figure 16 sensors-22-04251-f016:**
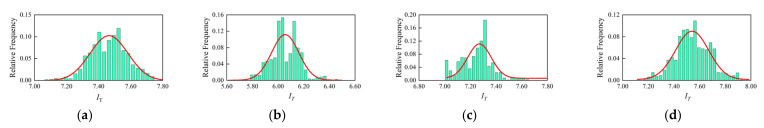
Posteriori relative frequency distribution of member 146. (**a**) Healthy; (**b**) failure of *x*-directional support restraint; (**c**) failure of *y*-directional support restraint; (**d**) failure of *z*-directional support restraint.

**Figure 17 sensors-22-04251-f017:**
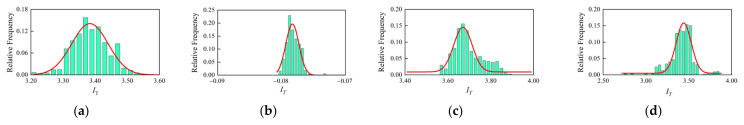
Posteriori relative frequency distribution of member 469. (**a**) Healthy; (**b**) failure of *x*-directional support restraint; (**c**) failure of *y*-directional support restraint; (**d**) failure of *z*-directional support restraint.

**Table 1 sensors-22-04251-t001:** Damage probability of each member under different damage conditions.

Damage Working Conditions	Rod Number		
	1	146	469
Mild injury	82.45%	100%	24.58%
Moderate damage	100%	100%	45.81%
Severe injury	100%	100%	99.79%

**Table 2 sensors-22-04251-t002:** Damage probability of each member under different bearing failure conditions.

Failure Condition of the Support Restraint	Rod Number		
	1	146	469
*x*-direction	100.00%	100.00%	100.00%
*y*-direction	100.00%	63.11%	100.00%
*z*-direction	2.80%	23.24%	34.51%

## Data Availability

Not applicable.

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
