# Peer review of "Damage Diagnosis of Single-Layer Latticed Shell Based on Temperature-Induced Strain under Bayesian Framework"

_sensors, 2022, doi:10.3390/s22114251_

Round 1

Reviewer 1 Report

I have the following comments:

- no citation of source materials for the formulas used,

- where does the measurement uncertainty value taken in the article come from - 0.2;

- there is no information in the applications regarding the continuation of the issues presented in the article. Do the authors anticipate further work related to this?

Author Response

We thank the reviewer for providing us the opportunity for some clarification during this reviewing process. Also, we find comments and suggestions provided by the reviewer are very helpful. Our responses to the comments and suggestions can be found in the attachment.

Reviewer 2 Report

The concept of the paper, especially the introduction is well written. Several comments:

- Please check the hyphen between letters on the paper, it's very hard to read.

- Please check the subscript letters in the maths and missing letters in all maths

- Probabilistic function on Page 7, equation 10 does not seem correct, same as the explanation in the following paragraph.

- Can the authors explain more in detail how the prior distribution was selected in this paper?

Author Response

(The authors gave the same response as above.)

Reviewer 3 Report

The paper is well written and scientific explanations are sound and adequate.  

Author Response

We are very grateful for your review of the paper and thanks very much for your recognition of our work.